# CAF-Released Exosomal miR-20a-5p Facilitates HCC Progression via the LIMA1-Mediated β-Catenin Pathway

**DOI:** 10.3390/cells11233857

**Published:** 2022-11-30

**Authors:** Yong Qi, Haibo Wang, Qikun Zhang, Zhiqiang Liu, Tianbing Wang, Zhengsheng Wu, Wenyong Wu

**Affiliations:** 1Department of General Surgery, The First Affiliated Hospital of Anhui Medical University, Hefei 230022, China; 2Department of Graduate School, Anhui Medical University, Hefei 230032, China; 3Department of General Surgery, Anhui No. 2 Provinicial People’s Hospital, Hefei 230011, China; 4Department of Pathology, School of Basic Medical Sciences, Anhui Medical University, Hefei 230032, China

**Keywords:** hepatocellular cancer, exosomes, LIMA1, miR-20a-5p, CAFs

## Abstract

Currently, exosomes derived from Cancer-associated fibroblast (CAF) have reportedly been involved in regulating hepatocellular carcinoma (HCC) tumour microenvironment (TME). LIM domain and actin binding 1 (LIMA1) is an actin-binding protein that is involved in controlling the biological behaviour and progression of specific solid tumours. We aimed to determine the effect of LIMA1 and exosome-associated miR-20a-5p in HCC development. LIMA1 and miR-20a-5p expression levels were examined by real-time quantitative PCR (qRT-PCR), western blotting or immunohistochemistry (IHC). Functional experiments, including Cell Counting Kit-8 (CCK-8), 5-ethynyl-2′-deoxyuridine (EdU) assays, colony formation assays, wound healing assays, and Transwell invasion assays, were performed to investigate the effect of LIMA1 and miR-20a-5p. A dual-luciferase reporter gene assay was performed to confirm the interaction of miR-20a-5p and LIMA1. Exosomes were characterised by transmission electron microscopy (TEM), nanoparticle tracking analysis (NTA), and western blotting. We noted that LIMA1 was downregulated in human HCC tissues and cells and remarkably correlated with overall survival (OS) and recurrence-free survival (RFS). LIMA1 overexpression suppressed HCC cell proliferation and metastasis in vitro and in vivo, while LIMA1 knockdown had the opposite effects. A mechanistic investigation showed that LIMA1 inhibited the Wnt/β-catenin signalling pathway by binding to BMI1 and inducing its destabilisation. Additionally, we found that LIMA1 expression in HCC cells could be suppressed by transferring CAF-derived exosomes harbouring oncogenic miR-20a-5p. In summary, LIMA1 is a tumour suppressor that inhibits the Wnt/β-catenin signalling pathway and is downregulated by CAF-derived exosomes carrying oncogenic miR-20a-5p in HCC.

## 1. Introduction

As one of the most frequently occurring cancers in the world, hepatocellular carcinoma (HCC) apparently increases the global cancer burden [1]. Although the incidence and mortality of HCC declined in China and Japan, it has been increasing in North America and in some European regions in recent years [2,3]. Viruses, including hepatitis B virus (HBV) and hepatitis C virus (HCV) infection, alcohol abuse and metabolic syndrome, are typical risk factors [3]. Despite tremendous advances in diagnosis and treatment, the inability to detect HCC earlier and prevent cancer recurrence remains a great challenge, and most patients with HCC ultimately die of multiple organ metastases [4]. Therefore, it is critical to address these problems.

LIM domain and actin binding 1 (LIMA1), also known as epithelial protein lost in neoplasm (EPLIN), encodes two isoforms, including a shorter Lima1-α (600 amino acids) and a longer Lima1-β (759 amino acids) isoform, both of which contain a central LIM domain and two actin-binding domains [5]. At the protein level, LIMA1 is an actin-binding protein that is considered to be a tumour suppressor over multiple ranges of malignancies. For instance, P53 binds to the response elements in the LIMA1 gene and induces LIMA1 expression, causing the suppression of tumour invasion [6]. Higher LIMA1 expression increased patient responsiveness to neoadjuvant chemotherapy (NAC), which greatly improved patient survival [7]. Apart from its tumour-suppressing function, LIMA1 also exerts effects on the pluripotency control of membrane dynamics and cellular metabolism [8]. However, the role of LIMA1 in HCC remained unclear.

The interaction between tumours and multiple mesenchymal components surrounding tumours leads to the development of complex tissue environments (tumour microenvironment, TME) that influence malignant tumour progression, including tumour growth, angiogenesis, metastasis, and chemoresistance [9]. As the main source of stromal cells, cancer-associated fibroblasts (CAFs) modulate malignant tumour phenotypes through a variety of strategies, such as extracellular matrix (ECM) remodelling, soluble growth factor secretion and signal transduction [10,11,12]. More recently, great attention has been given to the novel regulatory mechanism by which CAF-secreted exosomes regulate tumour progression [11,13].

Exosomes, a type of 50–150 nm membrane-bound nanovesicle mediating cellular communication among different types of cells through exocytosis, play a vital role in physiological and pathological conditions [14,15]. Exosomes have initially been deemed a vehicle for detaching useless cellular metabolites and can be secreted by almost all types of cells [16]. The components loaded in exosomes include lipids, proteins, and nucleic acids. Upon exocytosis by cells, exosomes fuse with the targeted cell membrane and then release the loaded contents, part of which might be involved in inter-cellular crosstalk and biological behaviours regulation in recipient cells. For example, B lymphoblastoid cell-secreted exosomes contain MHC class II-induced immune responses in vivo [17]. Cytoplasmic DNA secreted from cell exosomes augmented the beneficial effects on cell survival and homeostasis [18]. Moreover, exosomes have been reported to affect the pre-metastatic niche in cancer [19]. Recently, most researchers in this field have focused on the crosstalk between tumour cells and stromal cells mediated by exosome-loaded miRNAs [20]. MiR-20a-5p has been shown to be an oncogenic small noncoding RNA in breast cancer and cervical cancer and could be harboured in exosomes to regulate the TME and acute tubular injury [21,22,23,24]. However, whether exosome-associated miR-20a-5p is involved in HCC progression has not been well investigated.

In this study, we intended to determine the role of LIMA1 and exosome-associated miR-20a-5p in HCC progression. We found that LIMA1 acts as a tumour suppressor gene that predicts a better OS for patients and hinders HCC development via the Wnt/β-catenin signalling pathway. Additionally, decreased LIMA1 levels were mediated by CAF-secreted exosomes carrying miR-20a-5p, which enhanced HCC malignant phenotypes. Overall, our work suggests that some small molecules designed to target CAFs or activate LIMA1 expression could be a new treatment approach for HCC patients.

## 2. Materials and Methods

### 2.1. Tissue Specimens

HCC tissues were collected from ninety-two surgical patients who were clinically and histopathologically diagnosed from September 2013 to August 2015 at the First Affiliated Hospital of Anhui Medical University. Patients or their relatives signed a written informed consent form. The study was performed with the permission of the Ethics Committee of the First Affiliated Hospital of Anhui Medical University (Hefei, China).

### 2.2. Cell Culture and Transfection

We obtained human HCC cell lines (SMMC7721, Huh7, YY8103, Hep3B, Focus, HepG2 and HCCLM3) and a normal liver cell line (MIHA, WRL68) from the American Type Culture Collection (ATCC, Manassas, VA, USA). A 37 °C incubator with 5% CO2 was used to culture cells in Dulbecco’s modified Eagle’s medium (DMEM, Gibco, New York, NY, USA) containing 10% FBS (Biological Industries, Beit HaEmek, Israel) and 100 units/mL penicillin/streptomycin (Gibco, New York, NY, USA). Plasmids for LIMA1 overexpression or shRNA for LIMA1 knockdown were constructed as previously described [25].

### 2.3. RNA Isolation and Quantitative RT-PCR

The total RNA from HCC tissues and cell lines was extracted using a Fast Pure Cell/Tissue Total RNA Isolation Kit (Vazyme, Nanjing, China) according to the manufacturer’s protocols. Then, we reverse-transcribed the RNA to cDNA under standard conditions with a HiScript II 1st Strand cDNA Synthesis Kit (Vazyme, Nanjing, China). Real-time qPCR was performed using an SYBR Green PCR kit (Vazyme, Nanjing, China). We selected GAPDH as an internal control. The relative expression level of target genes was calculated as 2-ΔCt. The relative expression level of target genes in cancer tissues divided by the relative expression level of target genes in the matched adjacent para-tumoral tissues was higher than 1 in the high expression group and less than or equal to 1 in the low expression group. The primers of these targeted genes are listed as follows:

LIMA1: F, GACTCCCAGGTTAAGAGTGAGG

R, TTGCAGGTGCCTGAAACTTCT

CCND1: F, GCTGCGAAGTGGAAACCATC

R, CCTCCTTCTGCACACATTTGAA

CD44: F, CTGCCGCTTTGCAGGTGTA

R, CATTGTGGGCAAGGTGCTATT

JUN: F, TCCAAGTGCCGAAAAAGGAAG

R, CGAGTTCTGAGCTTTCAAGGT

TCF1: F, AACACCTCAACAAGGGCACTC

R, CCCCACTTGAAACGGTTCCT

BMI1: F, CCACCTGATGTGTGTGCTTTG

R, TTCAGTAGTGGTCTGGTCTTGT

### 2.4. Western Blotting

RIPA (Beyotime, Shanghai, China) containing 1% phenylmethanesulfonyl fluoride (PMSF) (Beyotime, Shanghai, China) was used to extract proteins from the HCC cells and tissues. The protein concentration was quantified using a bicinchoninic acid (BCA, Singapore) kit (Beyotime, Shanghai, China). For western blotting analysis, the proteins underwent separation by SDS-PAGE, nitrocellulose membrane transfer, quick blocker kit (Beyotime, Shanghai, China) blocking, and primary and secondary antibody incubation. After that, the bands were detected by ECL Plus (EMD Millipore, Billerica, MA, USA).

### 2.5. Coimmunoprecipitation (Co-IP)

Proteins were lysed with NP-40 (Beyotime, Shanghai, China) mixed with 1% PMSF. Then, a co-IP assay between LIMA1 and BMI1 was performed based on the Pierce™ Magnetic IP/Co-IP Kit (Thermo Scientific, Waltham, MA, USA). Subsequently, western blotting was performed to confirm the interaction between LIMA1 and β-catenin.

### 2.6. Immunohistochemistry (IHC) and Immunofluorescence (IF)

Slides of human tissues or mouse tumours were routinely deparaffinised and rehydrated, and antigen retrieval was performed in citrate buffer. Then, IHC analysis was performed using Super PlusTM High Sensitive and Rapid Immunohistochemical Kit (Elabscience, Wuhan, China) by following the manufacturer’s instructions. Images were obtained by light microscopy (Nikon, Tokyo, Japan). For IF analysis, the cells were fixed in 4% paraformaldehyde, washed with PBS, permeated with 0.5% Triton X and blocked with 10% donkey serum. Then, the cells were incubated with a primary antibody at 4 °C overnight, followed by a secondary antibody specific for IF. Images were obtained by fluorescence microscopy (Leica Microsystems Imaging Solutions, Cambridge, UK).

### 2.7. Functional Experiments

Cell functional experiments, including CCK-8, colony formation and Transwell assays, were performed as previously described [26]. A 5-ethynyl-20-deoxyuridine (EdU) incorporation assay was performed based on the instructions for the Cell-Light EdU DNA Cell Proliferation Kit (Beyotime, Shanghai, China).

### 2.8. CAF and NF Isolation

The isolation of CAFs from the HCC tissues and NFs in paired adjacent normal tissues was performed as described previously [27]. In brief, fresh tissues removed from patients of the First Affiliated Hospital of Anhui Medical University were kept in DMEM supplemented with 10% FBS. The tissues were then diced and digested with a mixture of collagenase and tryptase for 2 h at 37 °C in a shaking state. Then, the suspensions were resuspended in DMEM supplemented with 10% FBS and passed through 30 µm filters. After centrifugation, the pellets were collected, resuspended and cultured in the complete medium mentioned above. CAFs and NFs were identified based on morphology and immunofluorescence verification of the marker α-SMA.

### 2.9. Exosome-Associated Experiments

Exosomes were isolated from CAF- and NF-conditioned media supplemented with exosome-free serum after 48 h of culture. Then, conditioned media were collected and centrifuged at 500× *g* for 10 min and 10,000× *g* for 30 min at 4 °C. The supernatant was filtered through a 0.22 µm PVDF membrane (Millipore, Burlington, MA, USA) and ultracentrifuged at 110,000× *g* for 70 min at 4 °C. Exosomes obtained from the pellet were resuspended in PBS and preserved at −80 °C. Negative staining was used to identify exosomes by electron microscopy based on transmission electron microscopy (TEM). For the quantification analysis, a NanoSight NS300 instrument (Malvern Instruments Ltd. Malvern, UK) was used to track the size and density of the exosomes. For exosome labelling, the exosomes were labelled using PKH67 (Sigma, St. Louis, MO, USA). Labelled exosomes were obtained by repeating the step mentioned above. Two micrograms of exosomes were incubated with 5 × 105 recipient cells for 48 h.

### 2.10. Dual-Luciferase Reporter Assay

Plasmids containing wild-type (WT) and mutated (MUT) 3′-UTRs of LIMA1 were synthesised by Tsingke (Nanjing, China). WT and MUT LIMA1 3′-UTR plasmids were co-transfected with miR-20a-5p mimic into cells using Lipofectamine 3000 (Invitrogen, Waltham, MA, USA). The Dual-Luciferase Reporter Assay System (Promega, Madison, WI, USA) was used to determine the luciferase activity 48 h after transfection. We prepared three repeats for each sample, and the assay was repeated three times.

### 2.11. Mouse Models

We purchased six-week-old male nude mice (BALB/c-nu/nu mice) from the animal centre at Anhui Medical University and housed them in a pathogen-free environment. For tumorigenesis, 3 × 10^6^ YY8103 cells with a normal control vector or LIMA1-targeted vector were injected into the flank. The tumours in the mice were monitored and recorded every five days. Five weeks later, the tumours were removed from euthanised mice for further examination. For exosome-associated tumorigenesis, 3 × 10^6^ Huh7 cells were injected into nude mice flank. One week later, subcutaneous mouse xenografts were intratumorally injected with CAF-derived Exo/miR-20a-5p-mimic or Exo/NC-mimic. For the lung metastasis model, the 1 × 10^6^ YY8103 cells mentioned above from each group were injected into the tail vein. Eight weeks later, the mice were sacrificed, and the lungs were obtained for observation and further examination.

### 2.12. Statistics

Experiments were implemented at least three times, each of which was performed with three repeats. Data are presented as mean ± SD. GraphPad Prism 7 software (V7.04, Gpaphpad, San Diego, CA, USA) and SPSS Statistics 25 software (V25.0, IBM, Amunke, NY, USA) were used for statistical analysis. The data are expressed as the means ± SD, and Student’s *t*-test or the chi-square test was used for different determinations. The univariate and multivariate COX analysis was used to clear the prognostic impact of the potential prognostic factors. The linear relationship between miR-20a-5p and LIMA1 was analysed using Pearson’s correlation coefficient. The log-rank test was used to monitor the correlation between LIMA1 expression and survival. Significance was defined as *p* < 0.05.

## 3. Results

### 3.1. LIMA1 Was Downregulated in HCC and Positively Associated with Survival

To determine the expression of LIMA1 in HCC, we first examined its mRNA level in ninety-two pairs of HCC tissues by qRT-PCR. The results showed that the mRNA level of LIMA1 was markedly downregulated in cancer tissues compared to matched adjacent para-tumoral tissues (Figure 1A). Then, eight pairs of HCC tissues were selected to analyse the LIMA1 protein level. Compared with paired para-tumoral tissues, LIMA1 was evidently decreased in HCC tissues (Figure 1B). A consistent result was observed by IHC detection (Figure 1C). In addition, we found that LIMA1 expression was moderately decreased in patients with liver cirrhosis induced by HBV infection but decreased in HCC patients (Figure 1D), which suggests that LIMA1 expression might be associated with HCC progression. Further analysis showed that LIMA1 expression was apparently decreased in HCC cell lines (Figure 1E,F). These data showed that LIMA1 was frequently decreased in HCC and inversely associated with HCC progression.

Next, the LIMA1 expression level in cancer tissues divided by the LIMA1 expression level in matched adjacent para-tumoral tissues was higher than 1 in the high expression group (number = 30) and less than or equal to 1 in the low expression group (number = 62). Then, We analysed the clinical correlation of the LIMA1 expression level and the parameters of HCC patients between the high expression group(number = 30) and the low expression group(number = 62). The data indicated that LIMA1 was significantly associated with the HbsAg (*p* = 0.037), tumour size (*p* = 0.035) and tumour multiplicity (*p* = 0.039) (Table 1). Further analysis showed that HCC patients with higher LIMA1 expression levels had better overall survival (OS) and recurrence-free survival (RFS) (Figure 1G,H). Together, the data indicated that LIMA1 could be an excellent indicator for HCC patients. The univariate and multivariate COX analysis further suggested that the LIMA1 expression level was an independent risk factor for OS for patients with HCC (Table 2). Together, the data indicated that LIMA1 could be an independent factor for predicting the prognosis of HCC patients.

### 3.2. LIMA1 Suppressed HCC Cell Proliferation, Metastasis and EMT

To investigate the effects of LIMA1 on HCC progression, we transfected an overexpression plasmid into YY8103 and HCCLM3 cell lines with relatively lower LIMA1 levels and depleted the LIMA1 in the Huh7 cell line with relatively higher LIMA1 levels using a targeted shRNA (Appendix A). We found that LIMA1 overexpression evidently diminished cell viability and the capacity to form colonies, whereas LIMA1 silencing enhanced cell viability and colony formation (Figure 2A,B). The EdU assay showed that the percentage of proliferative cells was decreased after LIMA1 overexpression, while LIMA1 knockdown had the opposite effect (Figure 2C). In addition, transwell assays revealed that LIMA1 overexpression inhibited migration and invasion, while LIMA1 silencing enhanced migration and invasion (Figure 2D). Immunofluorescence and western blotting showed that the expression of the EMT marker vimentin was decreased in the LIMA1 overexpressing group but increased in the LIMA1 depleted group (Figure 2E). However, the expression level of E-cadherin, another EMT marker, showed the opposite alteration (Figure 2F). Together, our results demonstrate that LIMA1 might be a tumour suppressor that hinders HCC development in vitro.

### 3.3. LIMA1 Inhibited HCC Tumorigenesis In Vivo

Given the data that LIMA1 suppressed HCC progression in vitro, we next examined the role of LIMA1 in vivo. YY8103 cells transfected with LIMA1-overexpressing or negative control plasmids were injected into nude mice, and the tumours were recorded every week. Five weeks after inoculation, the mice were sacrificed, and their tumours were removed (Figure 3A). As displayed in Figure 3B, the growth of the tumours injected with LIMA1 overexpression cells was apparently lower than that of tumours injected with control cells. Consistent with this finding, the weight of tumours removed from mice in the overexpression group was smaller than those in the control group (Figure 3B,C). Subsequently, these removed tumours were verified by HE staining, and the IHC results showed that increased LIMA1 levels in vivo evidently abolished Ki67 and vimentin expression but facilitated E-cadherin expression (Figure 3D). Additionally, the metastatic nodes in the lung were visualised by the naked eye and verified by HE staining (Figure 3E). We found that the number of nodes was much lower in mice inoculated with LIMA1 overexpressing cells (Figure 3F). Collectively, these results showed that LIMA1 exerted tumour-suppressing effects on HCC in vivo.

### 3.4. LIMA1 Impaired the Wnt/β-Catenin Signalling Pathway through BMI1

To gain insight into the mechanism by which LIMA1 affects HCC development, we identified the protein partners that interact with LIMA1. By searching the online bioinformatic database Biogrid, Genemania and Hitpredict, we noted that AIFM1, BMI1, FER and PINK1 were the common interacting partners (Figure 4A). Co-IP and immunofluorescence analysis in Huh7 cells confirmed the endogenous interaction between LIMA1 and BMI1 rather than the other three (Figure 4B,C, Appendix A). Then, we found that BMI1, polycomb ring finger protein [28], was decreased in LIMA1-overexpressing YY8103 cells but increased in LIMA1-lacking Huh7 cells (Figure 4D). Considering that, we suspected that the dynamic BMI1 level with LIMA1 was affected by the binding of the LIMA1/BMI1 immunocomplex. In line with our hypothesis, we detected that elevated BMI1 in HCC tissues was not significantly correlated with LIMA1 in mRNA level (Appendix A), hinting that the effect of LIMA1 on BMI1 might be in protein level. Consistently, a study has reported that wild-type βTrCP binds to BMI1 and facilitates protein degradation in the ubiquitin-proteasome system (UPS) relied upon the way [29]. Given that, we first treated HCC cells with MG132, a regent for ubiquitination inhibition, and observed that the decreased BMI1 level was greatly reversed in the LIMA1 knockdown or overexpression group (Figure 4E). CHX treatment showed that BMI degraded rate was heavily enhanced when LIMA1 was upregulated but evidently decreased after LIMA1 depletion (Figure 4F). Subsequent IP analysis revealed that upregulated-LIMA1 increased BMI1 ubiquitination, whereas downregulated-LIMA1 exerted the opposite effect (Figure 4G). Further examination showed that LIMA1-mediated BMI1 degradation was K-48 ubiquitination-reliant manner (Figure 4H). Combined, the data suggested LIMA1 modulated BMI1 degradation via UPS.

Multiple studies have reported that BMI1 controls hematopoietic stem cell (HSC) self-renewal, induces epithelial cell proliferation and modulates malignant behaviours in various types of cancers, including hepatocellular carcinoma (HCC) via Wnt/β-catenin pathway [30,31,32]. We, therefore, tended to seek out whether LIMA1 exerts an effect on the Wnt/β-catenin pathway through BMI1. We detected weak alteration of β-catenin levels in HCC cells with increased or decreased LIMA1 levels (Appendix A). However, further subcellular location analysis showed that LIMA1-overexpressing cells increased the cytoplasmic location of β-catenin, while LIMA1 silenced cells maintained more β-catenin in the nucleus, as revealed by western blotting and immunofluorescence results (Figure 4I, Appendix A). Nuclear β-catenin acts to initiate the transcription of target genes. We observed that the downstream molecules, including CCND1, CD44, JUN and TCF1, were markedly decreased in LIMA1-overexpressing cells but increased in LIMA1-lacking cells (Figure 4J).

To further determine the role of BMI1 in LIMA1-mediated activation of the Wnt/β-catenin pathway, we depleted BMI1 in LIMA1-lacking Huh7 cells. We detected that BMI1 knockdown in LIMA1-lacking cells reduced nuclear β-catenin location and partly reversed the suppressive effects of LIMA1 on cellular proliferation, migration, invasion and EMT procession (Figure 4K–N). Together, our data indicated that the binding of LIMA1 to BMI1 abrogates activation of the Wnt/β-catenin pathway, thus suppressing β-catenin-mediated transcriptional activation.

### 3.5. MiR-20a-5p Is an Oncogene Targeting LIMA1

MiRNAs play a vital role in gene silencing [33]. To identify the miRNAs targeting LIMA1 expression, we combined miRNA-associated databases, including TargetScan, miRDB, miWALK and TCGA, and screened four significantly upregulated miRNAs (Figure 5A). Among those, miR-20a-5p was the only one that strongly upregulated in ninety-two HCC tissues and inversely correlated with the LIMA1 expression level (Figure 5B,C, Appendix A). Additionally, miR-20a-5p was increased in many other tumours based on the analysis of the TCGA database (Appendix A). Further exploration of the database showed that the area under the ROC curve (AUC) of miR-20a-5p was 0.75 in HCC (Appendix A). Besides, the AUC values were greater than 0.5 in various cancers, suggesting miR-20a-5p is an excellent indicator distinguishing cancer tissues and normal tissues (Appendix A).

Next, the miR-20a-5p expression level in cancer tissues divided by the miR-20a-5p expression level in matched adjacent para-tumoral tissues was higher than 1 in the high expression group (number = 52) and less than or equal to 1 in the low expression group (number = 40). We subsequently analysed the correlation between the miR-20a-5p expression level and clinical parameters in the high expression group and the low expression group and detected that miR-20a-5p expression level was significantly associated with tumour size and tumour stage (Table 3). Further analysis showed that HCC patients with higher miR-20a-5p expression levels had worse OS and RFS (Appendix A). Apart from that, the multivariate analysis further suggested that the miR-20a-5p expression level was an independent risk factor for RFS for patients with HCC (Table 2). Together, the data indicated that miR-20a-5p could be a promising factor for predicting the prognosis of HCC patients.

To further probe the relationship between miR-20a-5p and LIMA1, we identified the binding site between miR-20a-5p and LIMA1 using the TargetScan database and constructed the mutated LIMA1 3′-UTR (Figure 5D). A luciferase reporter assay showed that miR-20a-5p mimic transfection evidently decreased luciferase activity in the LIMA1-wild-type group but not in the LIMA1-mutation group (Figure 5E). The findings above showed that LIMA1 was targeted and inversely regulated by miR-20a-5p.

To obtain a deeper understanding of the miR-20a-5p functions in HCC, we depleted miR-20a-5p in YY8103 cell lines using a targeted inhibitor. Functional studies indicated that miR-20a-5p knockdown inhibited HCC cell proliferation, invasion, metastasis and EMT (Figure 5F–L). In addition, miR-20a-5p silencing increased the LIMA1 expression levels and cytoplastic β-catenin retention (Figure 5M). Additionally, the expression level of β-catenin downstream molecules decreased with miR-20a-5p depletion (Figure 5N). Overall, miR-20a-5p acts as an oncogene in HCC and negatively regulates LIMA1 expression.

### 3.6. LIMA1 Knockdown Restored the Oncogenic Role of miR-20a-5p

To verify the functional target of LIMA1 for miR-20a-5p, we performed rescue experiments. YY8103 cells were co-transfected with NC-inhibitor+sh-NC; miR-20a-5p-inhibitor+sh-NC; NC-inhibitor+sh-LIMA1 and miR-20a-5p-inhibitor+sh-LIMA1. LIMA1 knockdown in cells with miR-20a-5p inhibitor transfection partly rescued the proliferative effect of miR-20a-5p, as revealed by colony formation; and EdU assays (Appendix A). Similarly; transwell; wound healing, and immunofluorescence assays showed that LIMA1 silencing partially reversed the effects of miR-20a-5p on HCC cell invasion; metastasis; and EMT progression (Appendix A). Additionally, LIMA1 knockdown rescued part of the miR-20a-5p effects on the expression level of β-catenin downstream molecules (Appendix A). These results demonstrated that the oncogenic role of miR-20a-5p was partly mediated by LIMA1.

### 3.7. CAF-Derived Exosomes Carrying miR-20a-5p Facilitated HCC Cell Progression

Based on data from GSE113740 and GSE106817, we found that miR-20a-5p was one of the differentially upregulated circulating microRNAs in HCC patients (Figure 6A), which indicates that miR-20a-5p could be secreted into patients’ blood and might be involved in TME regulation. CAFs have been shown to be the main source of stromal cells inducing TME alterations. We, therefore, separated human CAFs from HCC tissues and normal fibroblasts (NFs) from paired normal tissues, both of whose typical characteristics were confirmed based on morphology and western blotting of specific markers (Figure 6B,C). Subsequently, we examined miR-20a-5p expression in normal hepatic cell lines, HCC cell lines, CAFs and NFs and observed a strikingly higher miR-20a-5p level in CAFs (Figure 6D). In addition, in situ hybridisation (ISH) further confirmed that miR-20a-5p was highly expressed in HCC stroma (Figure 6E). Exosomes are excellent extracellular vehicles that mediate cell-cell communication in the TME [14,15]. We speculated that higher miR-20a-5p levels could be transferred to HCC cells in the form of exosomes. By probing the online EVmiRNA database (http://bioinfo.life.hust.edu.cn/EVmiRNA#!//browse, accessed on 8 January 2019), we found that miR-20a-5p could be loaded into fibroblast-secreted exosomes (Figure 6F). Then, we incubated Huh7 cells with CAF cultures pre-treated with GW4869, an exosome-depleted kit, or not. The results showed that CAF culture-treated Huh7 cells significantly increased miR-20a-5p levels. However, the regulatory effect was abolished in Huh7 cells after incubation with CAF culture in the presence of GW4869 pre-treatment (Figure 6G). In addition, the percentage of proliferative Huh7 cells increased in the CAF-treated group but not in the GW4869-pre-treated group (Figure 6H). Thus, the CAF-secreted exosomes are involved in the regulatory effect of CAFs on HCC cells. Then, we separated and purified exosomes from CAFs and NFs, which were confirmed by western blotting of exosome markers and nanoparticle tracking analysis (NTA) (Figure 6I–K). Both CAF- and NF-derived exosomes were labelled with PKH-67 and then co-cultured with Huh7 cells for 12 h. As shown by fluorescence, we detected the absorption of PKH67-labelled exosomes into Huh7 cells and increased the level of miR-20a-5p in Huh7 cells (Figure 6L,M). We subsequently transfected an miR-20a-5p mimic labelled with CY3 (miR-20a-5p-CY3) into CAFs and then co-cultured them with Huh7 cells. After 48 h of incubation, we observed apparent Cy3 fluorescence in co-cultured Huh7 cells (Figure 6N). The results suggested that miR-20a-5p could be transferred from CAFs into HCC cells in a manner dependent on exosomes. In addition, the LIMA1 expression level was significantly decreased in Huh7 cells treated with exosomes obtained from CAFs transfected with miR-20a-5p mimic (exo-miR-20a-5p mimic) but increased in Huh7 cells treated with exosomes obtained from CAFs transfected with miR-20a-5p inhibitor (exo-miR-20a-5p inhibitor) (Figure 6O). In combining these data, we found that CAF-derived exosomes loaded with miR-20a-5p could be transferred into HCC cells and could suppress LIMA1 expression.

In vitro experiments were performed to study the influence of CAF-derived exosomes carrying miR-20a-5p on HCC cells. EdU, wound healing and Transwell assays showed that the exo-miR-20a-5p-mimic enhanced Huh7 cell proliferation, migration and invasion. By contrast, opposite effects were detected in the exo-miR-20a-5p-inhibitor group (Figure 7A–C). Subsequently, the in vivo effects of CAF-derived exosomes was determined in nude mice inoculated with Huh7 cells. A week inoculation later, mice were treated with PBS, exosomes derived from CAFs with miR-20a-5p transfection (Exo/miR-20a-5p-mimic) or not (Exo/NC-mimic). We detected that administration of CAF-originated exosomes significantly facilitated cancer growth when compared with mice treated with PBS, with an augmented effect observed when mice administrated Exo/miR-20a-5p-mimic as displayed in tumour volume and weight (Figure 7D–F). Together, we revealed that CAF-derived exosomes harbouring miR-20a-5p could facilitate HCC development.

## 4. Discussion

Although various technological advancements in diagnosis and treatment have been made in recent decades, improving the long-term survival of HCC patients remains a difficult challenge. Our study identified LIMA1 as a tumour-suppressing gene that hindered HCC progression and was positively associated with patient survival (OS and RFS). MiR-20a-5p acts as an oncogene in HCC and is harboured in CAF-derived exosomes that are transferred from CAFs to HCC cells, inhibiting LIMA1 expression. This action could be one of the mechanisms by which LIMA1 was downregulated in HCC tissues.

Our study first showed that LIMA1 was a tumour suppressor in HCC. However, the mRNA level of LIMA1 in HCC from the TCGA database was increased and inversely correlated with OS. We supposed that this correlation might be due to the distinguishing population composition and pathogenic factors between the data from TCGA and our centre HCC patients. TCGA databases were primarily American, including white (51.27%), Asian (43.91%), and black or African-American (4.82%) patients. In America, alcohol abuse and metabolism-associated liver diseases are the main pathogenic factors for HCC patients, while most patients who progress to HCC might have HBV-associated hepatitis and cirrhosis in China. Consistent with this finding, we showed that LIMA1 was also downregulated in HBV-induced cirrhosis compared with normal livers, which indicates that LIMA1 might be an HBeAg-associated factor.

Based on the online database, we identified the interaction between LIMA1 and BMI1. As a previous study showed that complex interactions might cause protein degradation in the UPS-dependent way [34], we subsequently demonstrated the degradation of BMI1 in protein level with LIMA1 was mediated by UPS. Considering that LIMA1 is an actin-binding protein rather than a member of E3 ligases, we remained unclear on how LIMA1 modulates BMI1 ubiquitination. Published research has indicated that β-TrCP is involved in BMI1 protein UPS-mediated degradation [29]. However, whether LIMA1 interferes with theβ-TrCP/BMI1 complex and subsequent degradation deserves to be further explored.

Given that LIMA1 was significantly decreased in HCC, we set out to determine the upstream molecules that exert effects on LIMA1. MiRNAs are crucial small RNAs involved in post-transcriptional regulation that modulate multiple aspects of cancer biology, such as proliferation, apoptosis, invasion/metastasis, and angiogenesis [35]. We first demonstrated that miR-20a-5p is an oncogene that silences LIMA1 expression in HCC cells. Subsequently, by probing GEO data, we observed that miR-20a-5p was differentially expressed in HCC patient blood, implying that miR-20a-5p might be a regulator in the HCC TME. We, therefore, separated CAFs, the main stromal component in the TME, and compared the level of miR-20a-5p with HCC cells. The results showed that the miR-20a-5p expression level in CAFs was evidently higher than that in HCC cells, which prompted us to speculate that miR-20a-5p in the HCC TME could be transferred between HCC cells and CAFs. Many studies have reported that CAF-derived exosomes carrying miRNAs induce crosstalk between different types of cells in the TME, contributing to cancer progression [11,36]. Then, we demonstrated that miR-20a-5p transfer from CAFs to HCC cells was mediated in the form of CAF-derived exosomes, which is consistent with previous findings.

In our study, we reported that CAFs exerted carcinogenic effects on HCC cells via the transfer of exosomes carrying miR-20a-5p. CAF-secreted exosomes affect tumour biological behaviours, while cancer-derived exosomes can activate CAFs [37]. However, CAFs are a group of functionally and molecularly heterogeneous fibroblasts originating from various cells, including endothelial cells (ECs), vascular smooth muscle cells and bone marrow-derived cells. Therefore, CAFs are equipped with multiple characteristics. Some CAFs, defined as cancer-restraining CAFs (rCAFs), play a tumour-suppressive role in bladder cancer and colon cancer [38,39]. However, the CAFs obtained in our study and many other studies were rarely purified into subpopulations partly due to limitations in separation techniques. Therefore, the characteristics and functions of the CAF subpopulation and the exosomes derived from the subpopulation merit further investigation.

## 5. Conclusions

In summary, our study identified LIMA1 as a novel tumour suppressor hindering HCC development. LIMA1 inhibits HCC cells by mediating the Wnt/β-catenin signalling pathway through BMI1. In addition, LIMA1 was targeted by CAF-derived exosomes carrying miR-20a-5p, causing LIMA1 silencing and cancer suppression in HCC. All these findings suggest that new treatment approaches around LIMA1 or targeting CAFs, exosomes and circulating miR-25a-5p might be helpful in HCC diagnosis and treatment.

## Figures and Tables

**Figure 1 cells-11-03857-f001:**
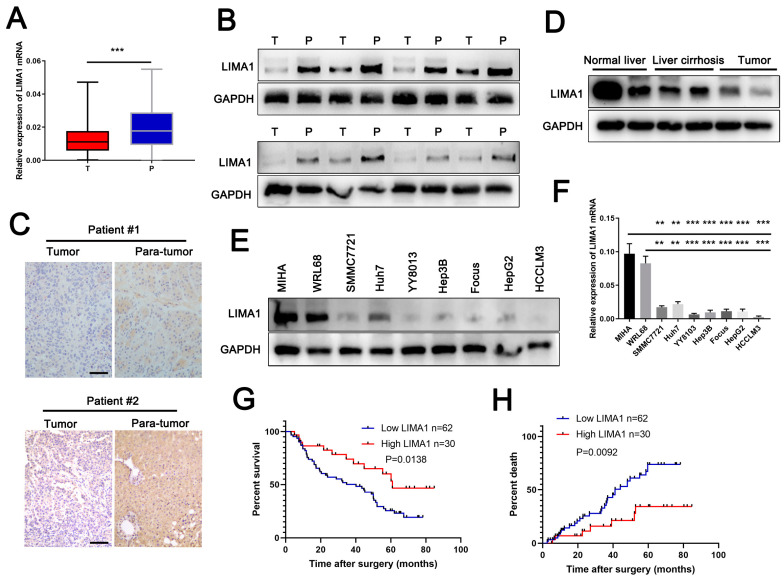
LIMA1 is downregulated in HCC tissues and cells. (**A**) RT-qPCR detection of LIMA1 mRNA expression in ninety-two pairs of HCC tissues. (**B**) Western blotting detection of LIMA1 protein expression in eight pairs of randomly selected HCC tissues. (**C**) IHC detection of LIMA1 protein expression in HCC tissues. (**D**) Western blotting detection of LIMA1 protein expression in normal liver, liver cirrhosis and HCC tissues. RT-qPCR (**E**) and western blotting (**F**) detection of LIMA1 expression in HCC cell lines and normal liver cells. (**G**) Overall survival (OS) and (**H**) recurrence-free survival (RFS) of HCC patients in the LIMA1-high and LIMA1-low groups. *** p* < 0.01, **** p* < 0.001.

**Figure 2 cells-11-03857-f002:**
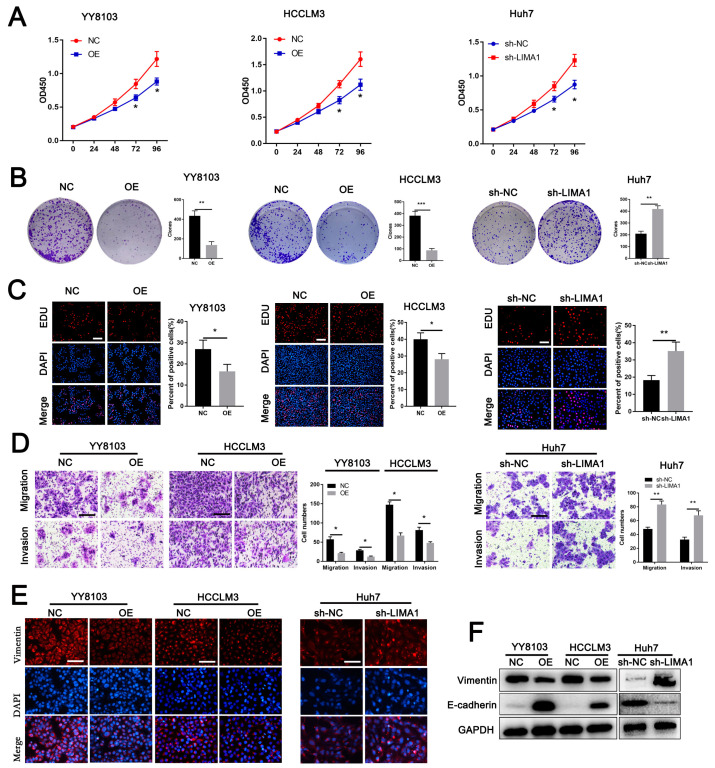
LIMA1 suppresses HCC cell malignant phenotypes in vitro. (**A**) CCK-8, (**B**) colony formation, and (**C**) EdU assays were performed to examine the effects of LIMA1 on proliferation. (**D**) Transwell assays were performed to examine the effects of LIMA1 on invasion and migration. Immunofluorescence for vimentin (**E**) and western blotting (**F**) for vimentin and E-cadherin were performed to detect the role of LIMA1 in EMT progression. (NC, normal control; OE, LIMA1 overexpression). ** p* < 0.05, ** *p* < 0.01, **** p* < 0.001.

**Figure 3 cells-11-03857-f003:**
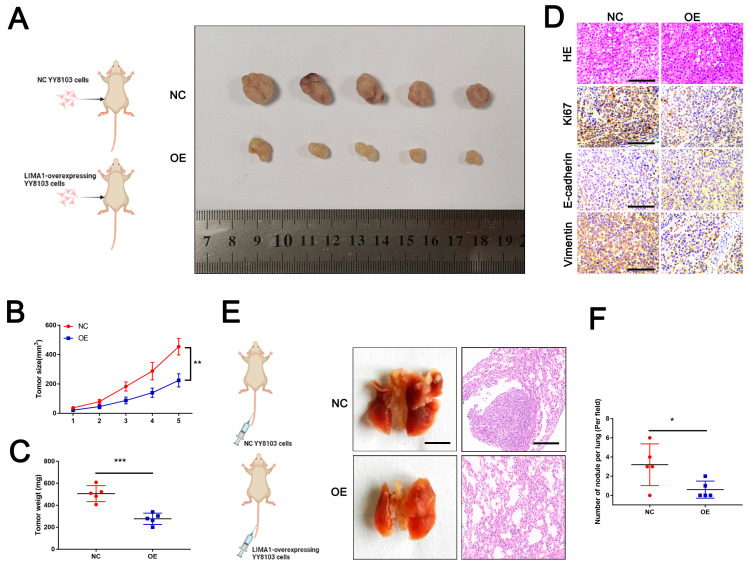
LIMA1 suppresses HCC progression in vivo. (**A**) The growth of tumours in the NC injection and LIMA1-OE groups was recorded every five days. (**B**) Tumours removed from mice injected with NC cells and LIMA1-OE cells after five weeks are displayed. (**C**) The tumour weight in the NC injection and LIMA1-OE groups was recorded. (**D**) HE staining and IHC for Ki67, E-cadherin and Vimentin were performed to detect the effects of LIMA1 on proliferation and EMT in vivo. The metastatic nodes in the lung were observed, confirmed by HE staining (**E**) and quantified (**F**). (NC, normal control; OE, LIMA1 overexpression). ** p* < 0.05, ** *p* < 0.01, *** *p* < 0.001.

**Figure 4 cells-11-03857-f004:**
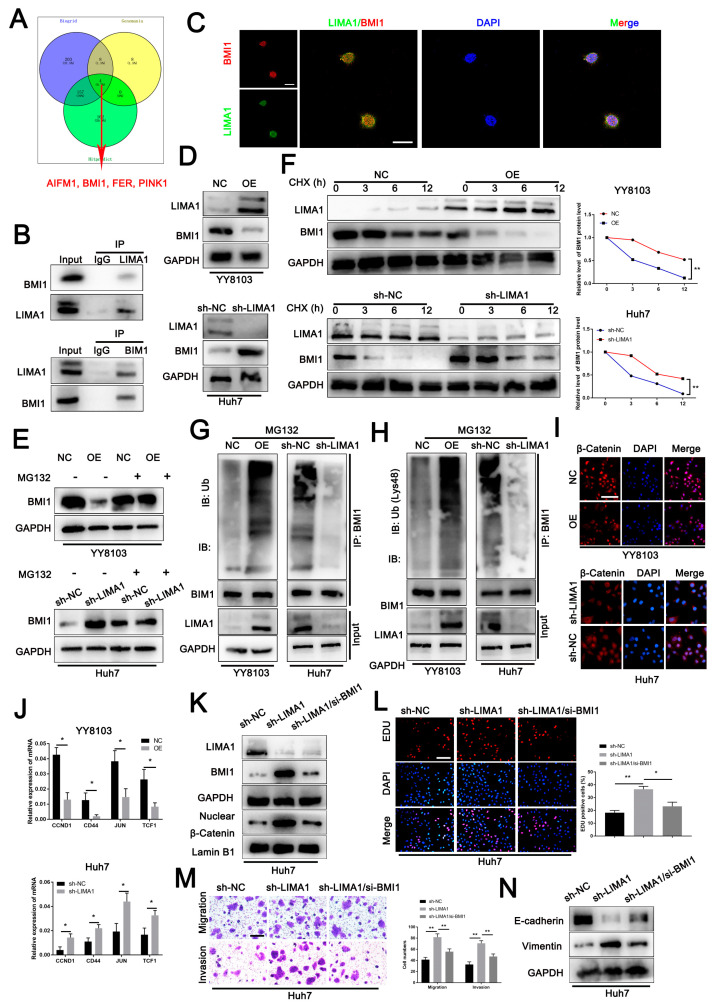
LIMA1 attenuated HCC progression by impairing the Wnt/β-catenin signalling pathway. (**A**) A-protein interaction network was predicted for LIMA1 based on Biogrid, Genemania and Hitpredict databases. AIFM1, BMI1, FER and PINK1 were identified as the common interacting partners for LIMA1. (**B**) Co-IP assay and (**C**) immunofluorescence were performed to verify the interaction between LIMA1 and BMI1. (**D**) BMI1 expression levels with LIMA1 knockdown or overexpression were examined by western blotting. (**E**) MG132 (20 μM) was used to pre-treat HCC cells with LIMA1 knockdown or overexpression for 8 h, and then BMI1 expression level was detected by western blotting. (**F**) HCC cells with LIMA1 knockdown or overexpression were treated with 100 μg/mL CHX, collected at the indicated times, and then subjected to IB with antibodies against BMI1. The relative BMI1 degraded rate was quantified relative to GAPDH. Cell lysates from HCC cells with LIMA1 knockdown or overexpression were subjected to IP with anti-BMI1, followed by IB with antibodies against (**G**) Ub and (**H**) Ub (Lys48). (**I**) Immunofluorescence was performed to detect the translocation of β-catenin with LIMA1 knockdown or overexpression. (**J**) The downstream molecules of β-catenin were examined by RT-qPCR. (**K**) BMI expression and nuclear β-catenin level were verified in Huh7 cells with BMI1 and LIMA1 silencing. (**L**) EDU, (**M**) transwell and (**N**) EMT markers examination were performed to detect the rescued effect of BMI1 on LIMA1. ** p* < 0.05, *** p* < 0.01.

**Figure 5 cells-11-03857-f005:**
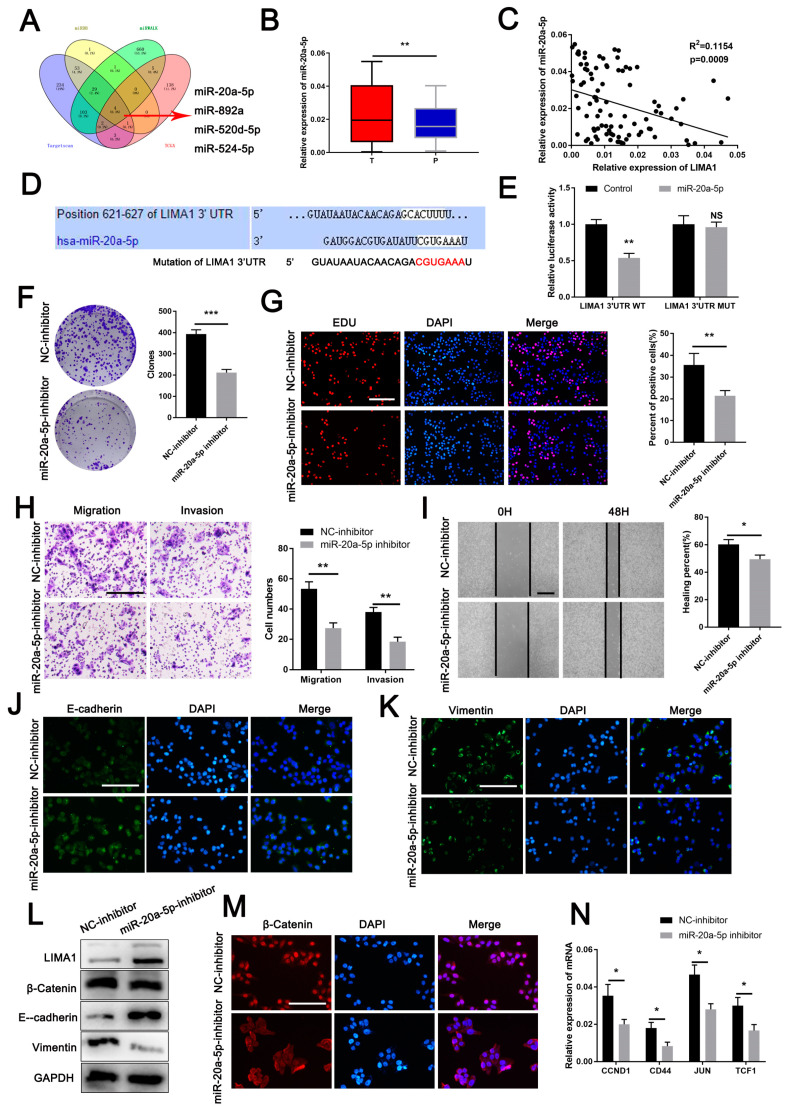
MiR-20a-5p is an oncogene targeting LIMA1. (**A**) The upstream miRNAs targeting LIMA1 were predicted based on the TargetScan, miRDB, miRALK and TCGA databases. (**B**) The expression level of miR-20a-5p in HCC tissues and (**C**) its correlation with LIMA1 were analysed. (**D**) The binding site between miR-20a-5p and LIMA1 was predicted by TargetScan, and the mutated site was constructed. (**E**) A dual luciferase reporter assay was performed to verify the target relationship between miR-20a-5p and LIMA1. (**F**) Colony formation and (**G**) EdU assays were performed to detect the effects of miR-20a-5p on proliferation. (**H**) Transwell and wound healing assays (**I**) were performed to examine the effects of miR-20a-5p on invasion and migration. Immunofluorescence for E-cadherin (**J**) and vimentin (**K**) and western blotting (**L**) for LIMA1, β-catenin, vimentin and E-cadherin were performed to detect the role of miR-20a-5p in EMT progression. (**M**,**N**) The translocation of β-catenin was examined by immunofluorescence, and the downstream molecules were examined by RT-qPCR. ** p* < 0.05, *** p* < 0.01.

**Figure 6 cells-11-03857-f006:**
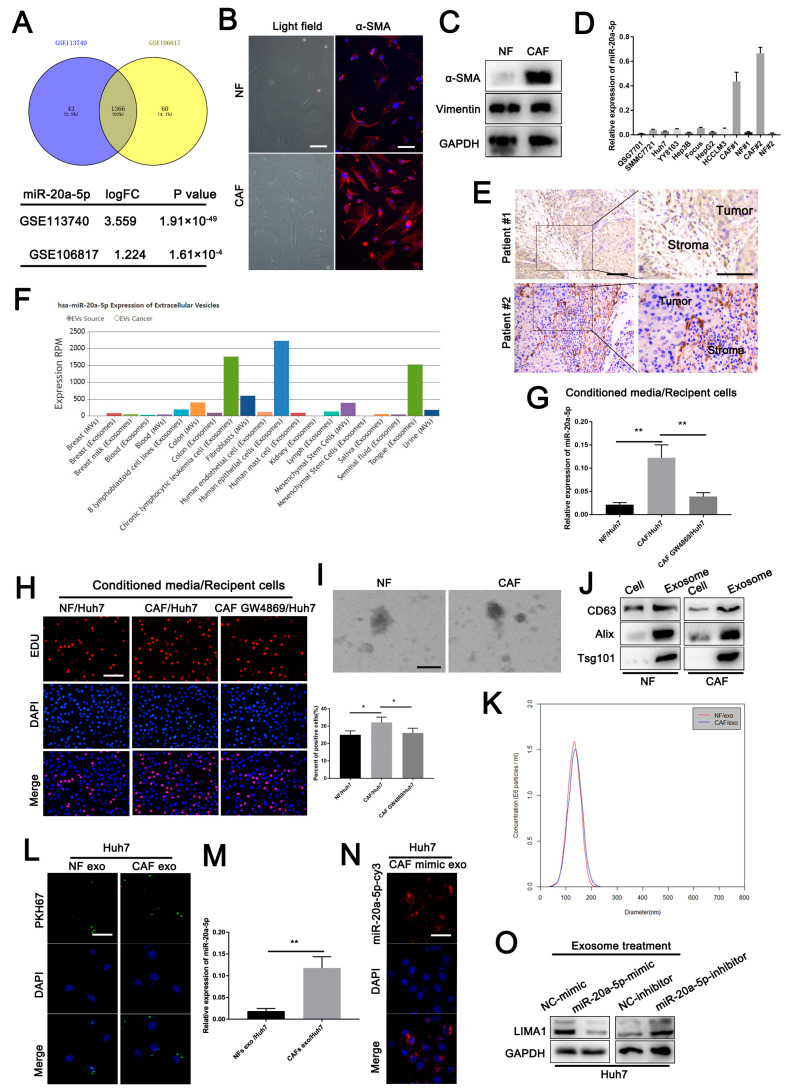
CAF-derived exosomes carrying miR-20a-5p-mediated HCC cell progression. (**A**) MiR-20a-5p was detected in HCC patients’ peripheral blood based on data sets GSE113740 and GSE106817. (**B**) CAFs and NFs obtained from two HCC patients were verified based on morphology and immunofluorescence for vimentin. (**C**) Western blotting for α-SMA and vimentin in CAFs and NFs was performed. (**D**) RT-qPCR was employed to examine the miR-20a-5p expression in normal liver cells, HCC cell lines, CAFs and NFs. (**E**) The location of miR-20a-5p in HCC tissues was determined based on in situ hybridisation (ISH). (**F**) The expression of miR-20a-5p in exosomes secreted from fibroblasts was predicted by using the EVmiRNA database. (**G**) Huh7 cells were cultured in NF and CAF media treated with GW4869 or not, and miR-20a-5p expression in Huh7 cells was detected by RT-qPCR. (**H**) Huh7 cells were cultured in NF and CAF media treated with GW4869 or not, and then an EdU assay was performed to examine the proliferation of Huh7 cells. Exosomes obtained from NF and CAF media were verified by electron microscopy (**I**), western blotting (**J**), and nanoparticle tracking analysis (**K**). (**L**) NFs and CAF-derived exosomes labelled with PKH67 were inoculated with Huh7 cells for 12 h, and then the uptake of the exosomes was detected by fluorescence confocal microscopy. (**M**) MiR-20a-5p expression in Huh7 cells treated with NFs and CAF-derived exosomes was examined by RT-qPCR. (**N**) Exosomes obtained from CAFs transfected with miR-20a-5p mimic labelled with CY3 were added to Huh7 cells, and the uptake of miR-20a-5p was detected by fluorescence confocal microscopy. (**O**) LIMA1 expression in Huh7 cells treated with exosomes obtained from CAFs transfected with miR-20a-5p mimic or inhibitor was examined by western blotting. ** p* < 0.05, *** p* < 0.01.

**Figure 7 cells-11-03857-f007:**
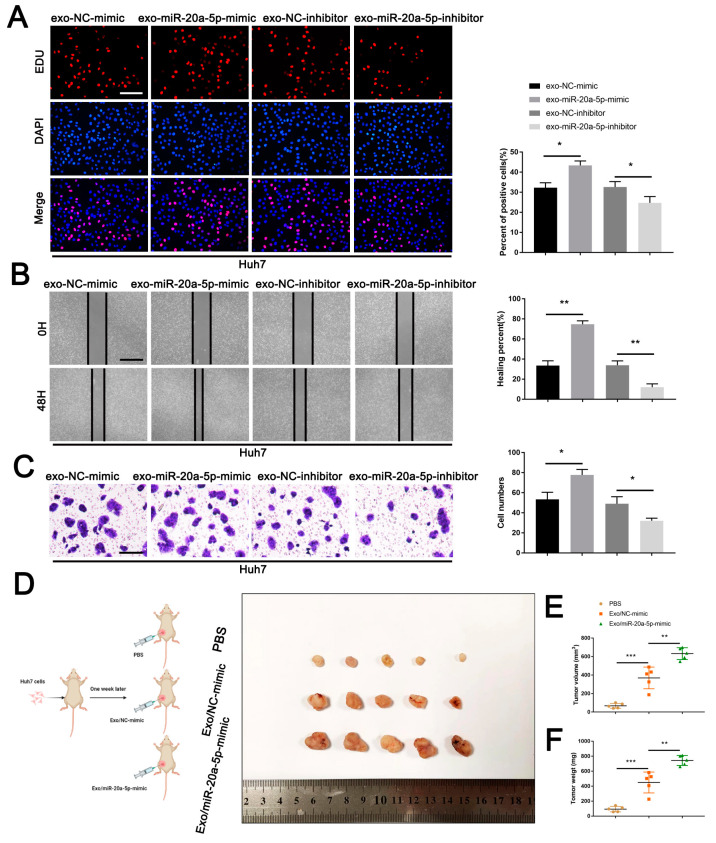
CAF-derived exosomes harbouring miR-20a-5p facilitated HCC progression. (**A**) EdU, (**B**) wound healing and (**C**) Transwell assays were performed to determine the effects of exosomes obtained from CAFs transfected with miR-20a-5p mimic or inhibitor on cell proliferation, migration and invasion. (**D**) Nude mice were divided into three groups inoculated with Huh7 cells. One week later, the tumours were treated with PBS or exosomes derived from CAFs with miR-20a-5p transfection (Exo/miR-20a-5p-mimic) or not (Exo/NC-mimic), then tumours were removed after 35 days inoculation. (**E**) Tumour volume and (**F**) weight were quantified. ** p* < 0.05, ** *p* < 0.01, *** *p* < 0.001.

**Table 1 cells-11-03857-t001:** The clinicopathological characteristics analysis of LIMA1 expression in HCC patients.

Features	LIMA1 Expression		No.	*p* Value
	Low (62)	High (30)		
**Age (Years)**				
≤60	32	15	47	0.885
>60	30	15	45	
Gender				
Male	37	18	55	0.976
Female	25	12	37	
HbsAg				
Positive	53	20	73	0.037 *
Negative	9	10	19	
Liver cirrhosis				
Yes	49	20	69	0.199
No	13	10	23	
AFP (ng/mL)				
>20	49	25	74	0.626
≤20	13	5	18	
Tumour size (cm)				
>5	50	18	68	0.035 *
≤5	12	12	24	
Tumour multiplicity				
Single	50	29	79	0.039 *
Multiple	12	1	13	
Stage				
I-II	24	13	37	0.672
III-IV	38	17	55	
Vascular invasion				
Yes	13	11	24	0.108
No	49	19	68	

Statistical analyses were performed using the chi-square test. * *p* < 0.05.

**Table 2 cells-11-03857-t002:** The univariate and multivariate COX analysis for the prognostic impact of LIMA1 and miR-20a-5p.

Parameters	OS			RFS
	*p* Value	HR (95% CI)	*p* Value	HR (95% CI)
Univariate Analysis				
Age (>60 vs. ≤60)	0.774	1.079 (0.641–1.817)	0.284	1.424 (0.746–2.717)
Gender (male vs. female)	0.488	1.210 (0.706–2.074)	0.236	1.501 (0.766–2.940)
Tumor number (single vs. multiple)	0.318	1.418 (0.714–2.815)	0.101	1.924 (0.880–4.208)
Tumor size (>5 cm vs. ≤5 cm)	0.001 *	3.180 (1.588–6.366)	0.002 *	4.046 (1.675–9.775)
AFP (>20 vs. ≤20)	0.012 *	2.646 (1.237–5.662)	0.067	2.273 (0.943–5.474)
HBsAg (positive vs. negative)	0.017 *	2.638 (1.185–5.869)	0.008 *	4.938 (1.506–16.187)
Liver cirrhosis	0.055	1.913 (0.987–3.707)	0.229	1.588 (0.747–3.378)
TNM stage	0.002 *	2.539 (1.420–4.540)	0.034 *	2.075 (1.056–4.079)
Vascular invasion (yes vs. no)	0.004 *	2.360 (1.310–4.254)	0.133	1.868 (0.827–4.221)
LIMA1	0.017 *	0.458 (0.242–0.868)	0.013 *	0.353 (0.155–0.803)
MiR-20a-5p	0.043 *	1.748 (1.019–3.000)	0.004 *	2.786 (1.379–5.629)
Multivariate analysis				
Tumor size (>5 cm vs. ≤5 cm)	0.035 *	2.151 (1.056–4.381)	0.005 *	3.620 (1.484–8.830)
TNM stage	0.013 *	2.126 (1.175–3.844)	-	-
Vascular invasion (yes vs. no)	0.020 *	2.042 (1.121–3.719)	-	-
LIMA1	0.032 *	0.488 (0.253–0.940)	-	-
MiR-20a-5p	-	-	0.014 *	2.425 (1.193–4.928)

Statistical analyses were performed using the univariate and multivariate COX analysis. * *p* < 0.05.

**Table 3 cells-11-03857-t003:** The clinicopathological characteristics analysis of miR-20a-5p expression in HCC patients.

Features	miR-20a-5p Expression		No.	*p* Value
	Low (40)	High (52)		
**Age (Years)**				
≤60	22	25	47	0.510
>60	18	27	45	
Gender				
Male	22	33	55	0.412
Female	18	19	37	
HbsAg				
Positive	28	45	73	0.052
Negative	12	7	19	
Liver cirrhosis				
Yes	27	42	69	0.145
No	13	10	23	
AFP (ng/mL)				
>20	29	45	74	0.092
≤20	11	7	18	
Tumour size (cm)				
>5	24	44	68	0.008*
≤5	16	8	24	
Tumour multiplicity				
Single	37	42	79	0.109
Multiple	3	10	13	
Stage				
I-II	21	16	37	0.035 *
III-IV	19	36	55	
Vascular invasion				
Yes	8	16	24	0.244
No	32	36	68	

Statistical analyses were performed using the chi-square test. * *p* < 0.05.

## Data Availability

The data that support the findings of this study are available from the corresponding author, W.-Y.W., upon reasonable request; in order to preserve patient privacy.

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
