# Peer review of "CAF-Released Exosomal miR-20a-5p Facilitates HCC Progression via the LIMA1-Mediated β-Catenin Pathway"

_cells, 2022, doi:10.3390/cells11233857_

Round 1

Reviewer 1 Report

In this paper, Qi et al. investigated the role of LIMA1 and miR-20a-5p in HCC progression. They demonstrated that LIMA1 expression was reduced in liver cancer tissues and cell lines and its downregulation/overexpression influenced cell growth, viability, invasion and migration in vitro and in vivo. They also demonstrated that LIMA1 exerted its effect by modulating Wnt/βCatenin pathway via BMI1 interaction and degradation. Finally, they demonstrated that the expression of LIMA1 was negatively regulated by miR-20a-5p, transferred into cancer cells by CAF-derived exosomes. The manuscript was interesting, with well conducted molecular studies.

There are some points still needed to be clarified.

- Figure 1A is very confusing. I would suggest to modify the way the authors expressed their results. They could show the mean or the median of the two groups or, if they want to maintain the link between the para-tumoral tissues and their corresponding cancers, they could change the colors or calculate a ratio for every pair of samples. Anyway, I think in the current way it is not clear.

- Figures 1E-F: could the authors add other normal liver cell lines? The difference between normal cells (just one cell line) and the cancer cell lines is not so clear especially in the western blot.

- Please, add the meaning of ‘NC’ and ‘OE’ abbreviations in supplementary Figure 1 and Figure 2 legends.

- Figure 2: why did not the authors try to obtain KO clones starting from Huh7 cell line (for example with CRISPR/Cas9 technology) to have a complete and stable depletion of the protein?

- Figures 2A-B: the colony-forming assay is more convincing than the growth curve experiment. Probably the colony assay was longer than the growth curve and the difference more appreciable. Could the authors comment on this and at least add the time duration of the colony-forming assay?

Figure 3: did the authors also perform the same experiments with Huh7 system and verify that LIMA1 silencing enhances tumor growth also in vivo?

Supplementary Figure 4:  see comments on Figure 1A.

Figure 5B: see comments on Figure 1A.

Figure 5E: did the authors expect a greater reduction in luciferase activity after miR-20a-5p exposure? Could they comment on this?

Paragraph 3.6 is not clear and I did not understand the aim of the experiments proposed. Could the authors clarify why they thought that the inhibition of the miRNA target LIMA 1 restored the oncogenic role of the miRNA, if also miR-20a-5p was inhibited itself?

Author Response

Dear Professor Reviewer 1,
Thank you very much for your comments. We have responded to each of your comments. Please see the attachment.

If you have any other comments, please feel free to contact us. Thank you very much again.

Kind regards.

Reviewer 2 Report

Qi et al. investigates the role of the protein LIMA1 in hepatocellular carcinoma (HCC). First, they demonstrate convincingly that in their patient derived HCC samples LIMA is downregulated. Via depletion and overexpression, they show that LIMA1 acts mainly as tumor suppressor. This has been done in several HCC cancer cell lines. Xenograft experiments support their conclusion.

At the molecular level they provide some evidence that LIMA1 influences BMI1 levels via ubiquitination pathways, which indirectly affects the WNT signaling pathway. Further they identified MiR-20a-5p as an upstream regulator of LIMA1 and they show that this miRNA can be transported via exosomes, which may contribute to the downregulation of LIMA in HCC.

Overall, I consider the work as solid, and convincing. Both the quantity and quality of the experiments justify publication in "Cells".

Major point:

1) In figures with error bars (e.g. Fig. 1F, 2A, etc.) it is unclear how many biological replicates were performed. This information should be included into a revised manuscript.

Some minor points:

1) In the sentence "In addition, we found that LIMA1 expression was moderately decreased in patients with liver cirrhosis induced by HBVs infection but significantly decreased in HCC patients" the authors should avoid the word "significantly", because no statistical test was performed.

2) In figue 1E,F QSG7701 are  endocervical adenocarcinoma (https://www.cellosaurus.org/CVCL_6944) and should not be declared as "normal liver cells". I think it is better to use the phrase "non-HCC" cell lines.

3) The presented "Full western" in the supplementary do not show full westerns. I am not sure whether this is ok for the journal.

4) In "Authors Contribution" "For research articles with several authors, a short paragraph specifying their

individual contributions must be provided. The following statements should be used" should be removed.

5) In the sentence " Higher EPLIN expression increased patient responsiveness to neoadjuvant chemotherapy" "EPLIN" should be replaced by "LIMA1".

Author Response

Dear Professor Reviewer 2,
Thank you very much for your comments. We have responded to each of your comments. Please see the attachment.

If you have any other comments, please feel free to contact us. Thank you very much again.

Kind regards.
